# TrkA Expression as a Novel Prognostic Biomarker in Oral Squamous Cell Carcinoma

**DOI:** 10.3390/ijms26146847

**Published:** 2025-07-16

**Authors:** Aleksandra Ciarka, Filip Skowronek, Przemysław Miłosz, Michał Kunc, Robert Burdach, Monika Sakowicz-Burkiewicz, Barbara Jereczek-Fossa, Anna Starzyńska, Rafał Pęksa

**Affiliations:** 1Department of Pathomorphology, Medical University of Gdansk, 7 Dębinki Street, 80-211 Gdańsk, Poland; aleksandra.ciarka@gumed.edu.pl (A.C.); michal.kunc@gumed.edu.pl (M.K.); 2Student Scientific Circle, Department of Cardiac Surgery, Medical University of Silesia, 15 Poniatowski Street, 40-055 Katowice, Poland; filipskrowronek2002@gmail.com; 3Student Scientific Circle, Department of Pathomorphology, Medical University of Gdansk, 7 Dębinki Street, 80-211 Gdańsk, Poland; przemyslaw.milosz@gumed.edu.pl; 4Department of Oral Surgery, Medical University of Gdansk, 7 Dębinki Street, 80-211 Gdansk, Poland; rburdach@gmail.com (R.B.); ast@gumed.edu.pl (A.S.); 5Department of Molecular Medicine, Medical University of Gdansk, 7 Dębinki Street, 80-211 Gdansk, Poland; ssak@gumed.edu.pl; 6Department of Oncology and Hemato-Oncology, University of Milan, 7 Festa del Perdono Street, 20-112 Milan, Italy; barbara.jereczek@ieo.i; 7Division of Radiotherapy, IEO European Institute of Oncology, IRCCS, 435 Ripamonti Street, 20-141 Milan, Italy; 8Department of Otolaryngology, Phoniatrics and Audiology, Collegium Medicum in Bydgoszcz, Nicolaus Copernicus University in Toruń, 13/15 Jagiellonian Street, 85-067 Bydgoszcz, Poland

**Keywords:** oral squamous cell carcinoma, *NTRK*, TrkA, prognosis, biomarker

## Abstract

Oral squamous cell carcinoma (OSCC) remains a significant global health challenge, representing 90% of oral malignancies. Despite therapeutic advances, patient outcomes remain poor, highlighting the need for novel prognostic biomarkers and treatment targets. We investigated the expression patterns of *NTRK* genes and their corresponding proteins (TrkA, TrkB, and TrkC) in OSCC, analyzing their relationships with clinical outcomes and potential as therapeutic targets. We examined 93 OSCC tissue samples using immunohistochemistry and quantitative real-time PCR. Protein expression was quantified using the H-score method. We analyzed correlations between Trk expression, clinicopathological parameters, and 2-year survival rates using chi-square tests, Mann–Whitney U tests, and Kaplan–Meier survival analysis. TrkA showed near-universal expression (97.8%—91 patients) in OSCC samples, with high expression levels significantly correlating with lower tumor grade (*p* = 0.014) and improved 2-year survival (*p* = 0.011). While TrkB and TrkC were expressed in 65.5% and 84.9% of cases, respectively, neither showed significant associations with clinical parameters. *NTRK2* and *NTRK3* mRNA levels demonstrated a strong positive correlation (R = 0.64, *p* = 0.002), suggesting coordinated regulation. Our findings establish TrkA as a promising positive prognostic marker in OSCC, warranting investigation as a therapeutic target. The strong correlation between *NTRK2* and *NTRK3* expression suggests shared regulatory mechanisms in OSCC pathogenesis. Further studies with larger cohorts and longer follow-up periods are needed to validate these findings and explore their therapeutic implications.

## 1. Introduction

Oral squamous cell carcinoma (OSCC), encompassing malignancies of the lip, floor of the mouth, tongue, gingiva, and buccal mucosa, represents approximately 90% of oral malignant neoplasms [1]. While other malignant tumors, such as mucoepidermoid carcinoma, adenoid cystic carcinoma, and myoepithelial carcinoma, occur in this region, they are considerably less frequent [2,3,4].

As the 16th most common malignant tumor worldwide, OSCC poses a significant global health challenge. The 2020 GLOBOCAN report documented 476,125 new cases and 225,900 deaths globally, highlighting its substantial public health impact [5].

The pathogenesis of OSCC involves multiple environmental and genetic factors. Key risk factors include tobacco use, alcohol consumption, betel quid chewing, viral infections, notably human papilloma virus (HPV), and Ebstein–Barr virus (EBV), and genetic predispositions, such as Fanconi anemia, xeroderma pigmentosum, and dyskeratosis congenita [6,7]. Tobacco use is a particularly significant risk factor, increasing the likelihood of developing oral cancer by approximately 8.4-fold, while users of smokeless tobacco face a 5- to 7-fold elevated risk [7]. Feghali et al. [8] identified smoking as an independent predictor of poor survival, alongside advanced age and higher pathological risk scores. Alcohol consumption synergistically amplifies tobacco’s carcinogenic effects, potentially increasing oral cancer risk up to 35-fold. Viral infections, particularly HPV16 and HPV18, contribute to approximately 6% of OSCC cases globally, predominantly in oropharyngeal tumors [9]. EBV infection, detected in 22.67–52% of cases, shows a notable association with buccal mucosa lesions in smokers [10,11,12]. OSCC has also been rarely reported in proximity to dental implants. While no direct causal relationship has been established, lesions in these areas are occasionally misdiagnosed as peri-implantitis, leading to delays in cancer diagnosis [13,14]. Ramos et al. [13] noted that OSCC near implants was more frequently observed in women without traditional risk factors such as smoking or alcohol consumption but with a history of leukoplakia.

The molecular pathogenesis of OSCC involves complex interactions of genetic and epigenetic alterations. Key mechanisms include activation of oncogenic pathways (PI3K/AKT/mTOR, MAPK, and Wnt-signaling) [7,15,16], DNA methylation changes, histone modifications, non-coding RNA dysregulation, and alterations in the tumor microenvironment and oral microbiome [16].

Prognostic factors in OSCC encompass various clinical and pathological parameters. Poor prognostic indicators include low socioeconomic status, diagnosis before age 40, single-modality treatment, advanced disease stage, lymph node metastases, tongue base localization, HPV-negative status, and positive surgical margins [17,18,19].

Adverse histological prognostic factors in OSCC include invasion depth > 5 mm, high tumor budding (poorly differentiated clusters or single cancer cells at the invasive front), perineural invasion, lymphovascular invasion, bone invasion, a multifocal invasive pattern (pattern 5), and a stromal-to-tumor ratio > 50% in deeply invasive tumors [2].

Molecular markers, including COX2, EGFR, cyclin D1, and p21 overexpression, have emerged as significant prognostic indicators [17].

The prognostic significance of programmed death-ligand 1 (PD-L1) depends on tumor location and the cell type expressing this immune checkpoint protein—either tumor cells or tumor-infiltrating lymphocytes [20]. High PD-L1 expression on tumor-infiltrating lymphocytes was associated with the N0 status and improved patient survival, regardless of tumor location. Conversely, PD-L1 expression on tumor cells appears to have variable effects on survival depending on the tumor’s location [20]. However, a meta-analysis by Nocini et al. [21] did not confirm the prognostic value of PD-L1 in OSCC, emphasizing challenges in standardizing its assessment.

*NTRK1*, *NTRK2*, and *NTRK3* (neurotrophic tropomyosin receptor kinase) refer to a group of genes that encode the TrkA, TrkB, and TrkC proteins, respectively, a family of receptor tyrosine kinases (RTKs). Trk receptors bind neurotrophins, regulating cellular processes such as proliferation, differentiation, metabolism, and apoptosis through phosphorylation of intracellular targets [22]. They play crucial roles in cellular regulation through neurotrophin binding and subsequent activation of MAPK, PI3K, and PKC pathways [23].

The most common oncogenic mechanism of Trk activation involves *NTRK* gene fusions, resulting from intra- or interchromosomal rearrangements. These fusions juxtapose the 3′ kinase domain-containing regions of *NTRK1*, *NTRK2*, or *NTRK3* with the 5′ region of another gene, creating a chimeric oncoprotein. This fusion protein leads to constitutive Trk activation, independent of ligand binding, driving uncontrolled signaling pathway activation and promoting oncogenesis [23,24].

Other mechanisms of Trk activation in cancer include *NTRK* mutations, alternative splicing variants, and Trk overexpression [23]. These alterations further underscore the pivotal role of *NTRK* genes in tumorigenesis and their potential as therapeutic targets.

Given the incomplete understanding of the etiopathogenesis of OSCC and the unsatisfactory treatment outcomes, the identification of new biomarkers and potential therapeutic targets remains a priority.

In this study, we additionally aim to compare the immunohistochemically assessed protein expression levels of *NTRK1*, *NTRK2*, and *NTRK3* genes in OSCC tissues and to analyze their associations with clinicopathological parameters, including age, sex, risk factors (alcohol consumption and tobacco use), tumor stage, treatment response, and two-year survival rates.

Furthermore, we intend to evaluate the correlation between the mRNA expression levels of the *NTRK1*, *NTRK2*, and *NTRK3* genes and these clinicopathological features.

## 2. Results

### 2.1. Expression of Trk Proteins and Clinicopathological Correlations

Assessment of Trk protein expression was carried applying the H-score approach, which integrates both the intensity of staining (graded on a scale from 0 to 3) and the proportion of positively stained cells (ranging from 0% to 100%). TrkA protein expression was detected in 91 cases (97.8%), with a mean H-score of 187 (median 205, range 0–300, IQR 115–260). Analysis revealed a significant correlation between TrkA expression and histological grade (***p* = 0.014**). However, when analyzed as a categorical variable (low/high expression), TrkA showed no significant associations with clinicopathological features apart from histological grade (***p* = 0.024**) (Table 1 and Table 2).

TrkB protein was expressed in 61 cases (65.5%), with a mean H-score of 33.5 (median 5, range 0–300, IQR 0–25). No significant correlations were found between TrkB expression and clinicopathological variables, whether analyzed as continuous or categorical data (Table 2 and Table 3).

TrkC protein expression was observed in 79 cases (84.9%), with a mean H-score of 0.833 (median 0, range 0–112.5, IQR 0). Analysis of TrkC expression as a categorical variable showed no significant associations with clinicopathological parameters, although a trend toward an association between TrkC expression and pN status was observed (*p* = 0.060) (Table 2).

### 2.2. Survival Analysis

High TrkA expression in primary oral cancer lesions correlated significantly with improved prognosis compared to cases with low expression (*p* = 0.011). Conversely, low expression of TrkB and TrkC showed a trend toward better prognosis, although these associations did not reach statistical significance (*p* = 0.150 and *p* = 0.300, respectively). Survival curves are presented in Figure 1.

### 2.3. NTRK Gene Expression Correlations

Analysis of *NTRK* mRNA levels revealed distinct correlation patterns. *NTRK1* mRNA level showed weak negative correlations with both *NTRK2* mRNA level (R = −0.12, *p* = 0.570) and *NTRK3* mRNA level (R = −0.06, *p* = 0.790), neither reaching statistical significance. In contrast, *NTRK2* and *NTRK3* demonstrated a strong positive correlation (R = 0.64, *p* = 0.002). Correlation plots with 95% confidence intervals are shown in Figure 2.

## 3. Discussion

The current literature provides limited data regarding the expression and clinical significance of *NTRK1/2/3* genes in OSCC. Our immunohistochemical analysis revealed differential expression patterns of Trk proteins, with TrkA, TrkB, and TrkC detected in 97.8%, 65.5%, and 84.9% of cases, respectively. High TrkA expression demonstrated a significant correlation with lower tumor grade (*p* = 0.014), while TrkB and TrkC showed no significant associations with clinicopathological parameters. Notably, patients with high TrkA expression exhibited significantly improved OS rates compared to those lacking TrkA expression (*p* = 0.011). In contrast, TrkB and TrkC expression showed trends towards poorer survival outcomes, although these associations did not reach statistical significance (*p* = 0.15 and *p* = 0.300, respectively). One possible explanation for the limited prognostic value of TrkB and TrkC in our study is their complex and context-dependent biological role. The literature suggests that both TrkB and TrkC can exhibit dual functions depending on the tumor type, stage, or presence of specific ligands. For instance, TrkB has been linked to aggressive behavior and chemoresistance in several cancers due to its role in promoting epithelial-mesenchymal transition (EMT) [25]. As for TrkC, its role is even more ambiguous. While some studies have identified TrkC as a potential tumor suppressor in the absence of its ligand neurotrophin-3 (NT3) [26], other reports show it may support tumor growth under different conditions [27]. This dichotomy might explain the lack of association with clinical outcomes in our cohort, particularly in a heterogeneous tumor like OSCC, where co-factors such as neurotrophin availability, receptor isoforms, and tumor microenvironmental context play a critical role.

A comparative Mexican study using PanTrk immunostaining found positive expression in only 42% of cases (n = 95) [28]. Direct comparison with our findings is limited due to the non-specific nature of PanTrk antibodies, which cannot distinguish between individual Trk proteins. Their analysis revealed age-dependent associations between PanTrk expression and tumor location, with higher prevalence in gingival tissues among patients over 60 years and lateral tongue among those under 60 years (*p* = 0.017). In contrast, our study found no correlation between TrkA expression and tumor location, identifying a significant association only with tumor grade (*p* = 0.014). Additional insights come from Escoto-Vasquez et al. [29], who reported differential expression patterns with high *NTRK2* mRNA levels in poorly differentiated OSCC and high *NTRK3* mRNA in well-differentiated cases, suggesting distinct roles for TrkB and TrkC in OSCC pathogenesis. However, their small sample size (n = 24) necessitates cautious interpretation of these findings.

Collectively, our findings support the role of Trk proteins in OSCC pathogenesis and highlight their potential as both prognostic biomarkers and therapeutic targets. Molecular analysis revealed a strong positive correlation between *NTRK2* and *NTRK3* mRNA expression (R = 0.64, *p* = 0.002). The strength and exclusivity of this correlation suggest potential shared regulatory mechanisms governing these genes in OSCC tissue. TrkA, a high-affinity receptor for nerve growth factor (NGF), is known to activate several key signaling cascades—most notably the PI3K/AKT, RAS/MAPK, and PLCγ pathways—resulting in enhanced cell survival, proliferation, and differentiation [30,31]. These pathways are frequently implicated in head and neck SCCs and could plausibly mediate the observed association between high TrkA expression and improved differentiation and survival in our cohort. We now explicitly state that further functional studies, including in vitro and in vivo models, are needed to confirm these mechanistic links and to clarify whether TrkA exerts a causal role in tumor suppression or reflects a more differentiated tumor phenotype [23].

Further functional studies are needed to explore these regulatory pathways and their implications for tumor biology.

*NTRK* gene fusions are predominantly associated with rare, aggressive malignancies, including infantile fibrosarcoma, secretory breast carcinoma, and salivary gland secretory carcinoma [32,33]. These genetic alterations occur less frequently in common cancers such as colorectal adenocarcinoma, lung adenocarcinoma, non-small cell lung cancer, and papillary thyroid carcinoma. However, emerging evidence demonstrates the significance of *NTRK* gene expression and Trk protein function across various squamous cell carcinomas (SCCs), including those of the oropharynx, esophagus, uterine cervix, and lung.

In their study of oropharyngeal cancer, Cho et al. [34] demonstrated a significant association between high TrkA expression and improved OS and recurrence-free survival (RFS). However, they observed contrasting results in OSCC, where PanTrk expression correlated with shorter RFS. Their research also suggested a potential mechanistic link between HPV infection and TrkA expression, as demonstrated by increased TrkA expression in SCC cell lines following HPV 16 E6/E7 gene transfer. These findings underline the importance of context-specific factors, such as HPV status, in interpreting Trk protein roles across SCC subtypes.

Studies of cervical malignancies provide additional insights into Trk protein expression patterns. Cervical SCC showed a higher prevalence of TrkA overexpression compared to cervical adenocarcinomas [35]. This pattern differed from our OSCC findings, as cervical SCC showed no association between TrkA expression and tumor grade, contrasting with the significant grade correlation we observed in OSCC. In esophageal SCC, Yu et al. [36] identified a distinct pattern, where increased *NTRK1* copy numbers correlated with poorer clinical outcomes, providing further evidence for the tissue-specific roles of *NTRK* alterations in different SCC types.

A comprehensive study by Terry et al. [37], analyzing TrkA and TrkB expression in 686 lung cancer cases, revealed distinctive patterns in SCC. They found TrkA or TrkB expression in over 90% of SCC cases, markedly higher than in other cancer types, where expression was sporadic. Furthermore, TrkB expression in lung SCC correlated with improved OS (*p* = 0.047) and disease-free survival (*p* < 0.001), paralleling our observations regarding the prognostic significance of Trk proteins in OSCC.

These findings have important clinical implications for OSCC management. The strong association between TrkA expression and improved survival suggests its potential use as a prognostic marker in clinical practice. Additionally, the tissue-specific expression patterns and functional roles of Trk proteins across different SCC types highlight the need for personalized therapeutic approaches. Overall, these findings emphasize the complexity and tissue-specific nature of *NTRK* gene alterations and Trk protein expression across SCCs. Further research is warranted to validate these results and explore the mechanisms underlying these associations, with the goal of advancing personalized therapeutic approaches in OSCC and related malignancies.

## 4. Materials and Methods

### 4.1. Study Population and Clinical Data

The approval of the independent bioethics committee was granted for conducting this study. The study cohort comprised 93 OSCC patients (31 women and 62 men) who underwent surgical treatment in the University Clinical Centre in Gdańsk, Poland. Patient survival data were obtained from clinical records and the Social Insurance Institution in Gdańsk database. The median follow-up period was 24 months, with a 2-year overall survival rate of 36.6%.

Comprehensive clinicopathological data were collected for each patient, including demographic factors, disease characteristics (tumor stage according to the 8th TNM classification, histological grade, tumor location), treatment modalities, and risk factors such as alcohol consumption and smoking status. Mortality data were also recorded throughout the follow-up period. All patient characteristics are summarized in Table 3.

### 4.2. Tissue Processing and Immunohistochemistry

Fresh tumor specimens were fixed in 10% formalin for 48 h, followed by dehydration and paraffin embedding. Hematoxylin and eosin-stained sections were reviewed to confirm diagnosis and disease staging. Representative tissue areas were selected based on typical tumor morphology, preserved architecture, and absence of necrosis, thermal damage, or significant inflammatory infiltrates.

Tissue microarrays (TMAs) were constructed using a Manual Tissue Arrayer I (Beecher Instruments, MTAI, K7 BioSystems) with 1.5 mm diameter cores. Each tumor was represented by at least two tissue cores, resulting in seven microarray blocks. Control tissues (palatine tonsils and stomach) were included in each block for orientation. Immunohistochemical analysis was performed using specific monoclonal antibodies (Cell Signaling, Danvers, MA, USA: TrkA 12G8, TrkB 80G2, TrkC C44H5) following the manufacturer’s protocols. Two pathologists (AC and RP) independently evaluated receptor expression using the H-score method, which combines staining intensity (0–3) and percentage of positive cells (0–100%), using an Olympus BX43 microscope (Tokyo, Japan) (Figure 3).

### 4.3. Molecular Analysis

From paraffin blocks of 30 patients with oral cancer, 10 μm × 2 μm sections were cut, which were then subjected to deparaffinization and RNA extraction. Total RNA isolation from FFPE sections was performed using a commercial kit, the RNeasy FFPE Mini Kit (QIAGEN, Hilden, Germany), following the manufacturer’s instructions.

The concentration of the isolated RNA was determined using fluorometric methods with a Qubit fluorometer (Invitrogen, Waltham, MA, USA) and the commercial Qubit RNA HS assay kit, in accordance with the manufacturer’s protocols.

The mRNA levels of the *NTRK1*, *NTRK2*, and *NTRK3* genes were determined relative to the transcript level of the reference gene beta-actin (ACTB) using real-time RT-qPCR with TaqMan probes. The RT-qPCR reactions were carried out using the Path-ID™ Multiplex One-Step RT-PCR Kit (Applied Biosystems, Foster City, CA, USA) and Universal ProbeLibrary Probes for Human (Roche) on the LightCycler 480 II system (Roche), with specific primers for both the target and reference transcripts. Each measurement was performed in triplicate, and the final result was reported as the mean of these replicates. The reproducibility of the obtained results was assessed based on the standard deviation and coefficient of variation for each sample. Each RT-PCR reaction included control samples: a negative control (no RNA template) and a positive control, to exclude contamination and confirm the efficiency of the reaction.

The relative expression level of each transcription factor gene was presented as a normalized ratio of the expression level of the target transcript to that of the reference gene transcript. Data were processed using LightCycler® 480 II Software, version 2.0 (Roche Diagnostics, Basel, Switzerland).

### 4.4. Statistical Analysis

Data analysis was performed using Statistica 13 (TIBCO Software Inc., Palo Alto, CA, USA) and the R programming environment, version 4.3.2. The chi-square test was used for categorical variables, with Fisher’s exact test applied to small sample sizes. The Mann–Whitney U test was employed for quantitative variables due to non-normal distribution. Survival analysis utilized the Kaplan–Meier method with the log-rank test for curve comparison. Statistical significance was set at *p* < 0.05.

## 5. Conclusions

Our comprehensive analysis of *NTRK1*, *NTRK2*, and *NTRK3* expression in OSCC reveals distinct patterns with potential clinical implications. High TrkA protein expression emerged as a favorable prognostic indicator, significantly correlating with lower tumor grade and improved survival outcomes. While elevated TrkB and TrkC expression showed trends toward poorer outcomes, these associations require validation through larger studies.

The strong positive correlation between *NTRK2* and *NTRK3* mRNA levels suggests shared regulatory mechanisms in OSCC pathogenesis, opening new avenues for molecular research. These findings collectively support the role of Trk proteins in OSCC development and highlight their potential as both prognostic biomarkers and therapeutic targets, with TrkA particularly promising for clinical applications.

### Limitations of This Study

Several important limitations should be considered when interpreting our findings. First, while our cohort of 93 patients provided sufficient statistical power for primary analyses, the sample size and single-center nature of this study may limit the generalizability of the results to broader OSCC populations. An additional limitation regarding the significance of Trk protein expression in our OSCC cohort is the lack of HPV and EBV status assessment in the cancer cells, as these factors can influence tumor biology and potentially affect Trk expression patterns.

Our analysis focused on Trk proteins and mRNA expression levels without exploring functional aspects of *NTRK* gene alterations, such as fusions, mutations, or splicing variants. This limitation prevents definitive conclusions about the mechanistic role of these genes in OSCC pathogenesis. Similarly, while we identified a significant correlation between *NTRK2* and *NTRK3* mRNA levels, the underlying regulatory mechanisms remain unexplored due to a lack of functional validation studies. Furthermore, this study did not include data on HPV and EBV status in cancer cells, which may have influenced survival outcomes and clinicopathological factors. Additionally, despite the use of standardized protocols, the inherent subjectivity of immunohistochemical H-score evaluation could compromise the reproducibility of the results.

Although we discussed comparisons with SCCs from other anatomical sites, the absence of direct comparative analysis between OSCC and other SCC subtypes limits our ability to establish tissue-specific patterns of *NTRK* expression.

Future research addressing these limitations through larger multicenter studies, extended follow-up periods, functional analyses, and direct comparative studies will enhance our understanding of *NTRK* genes’ role in OSCC and strengthen the clinical applicability of these findings.

## Figures and Tables

**Figure 1 ijms-26-06847-f001:**
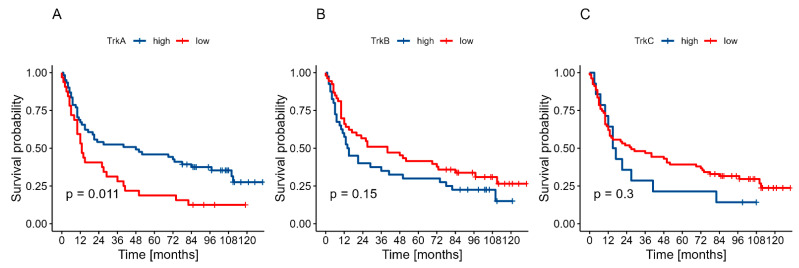
Two-year OS in relation to expression levels: (**A**)—TrkA, (**B**)—TrkB, (**C**)—TrkC.

**Figure 2 ijms-26-06847-f002:**
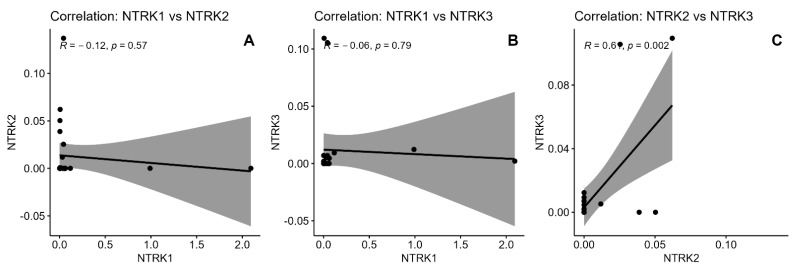
Graphical correlation between mRNA levels of *NTRK* genes. (**A**)—Correlation: *NTRK1* vs. *NTRK2*, (**B**)—Correlation: *NTRK1* vs. *NTRK3*, (**C**)—Correlation: *NTRK1* vs. *NTRK2*.

**Figure 3 ijms-26-06847-f003:**
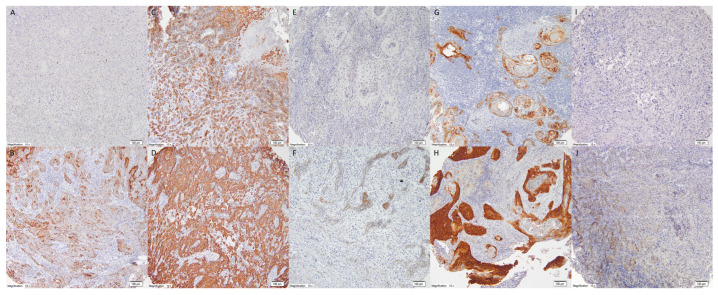
Expression of TrkA in OSCC—membranous and cytoplasmic staining intensity according to H-score: (**A**). 0, (**B**). 1, (**C**). 2, (**D**). 3, expression of TrkB in OSCC—membranous and cytoplasmic staining intensity according to H-score: (**E**). 0, (**F**). 1, (**G**). 2, (**H**). 3, expression of TrkC in OSCC—membranous and cytoplasmic staining intensity according to H-score: (**I**). 0, (**J**). 1. (magnification: 10×).

**Table 1 ijms-26-06847-t001:** TrkA, TrkB, and TrkC expression (low vs. high) in relation to clinicopathological features. *p*-values were calculated using the chi-square test.

Feature		TrkA High	TrkA Low	*p*	TrkB High	TrkB Low	*p*	TrkC High	TrkC Low	*p*
Gender	Female	23	8	0.217	14	17	0.767	4	27	0.682
	Male	38	24		26	36		10	52	
Grade	High	31	24	**0.024**	24	31	0.883	10	45	0.310
	Low	30	8		16	22		4	34	
Stage	High	28	16	0.507	19	25	0.502	6	38	0.321
	Low	22	9		11	20		2	29	
pN	0–1	29	9	0.112	13	21	0.902	5	29	0.060
	2–3	20	14		14	24		1	37	
pT	1–2	29	11	0.325	15	19	0.423	5	29	0.319
	3–4	21	13		14	26		3	37	
Localization	Other	24	14	0.681	14	24	0.317	7	31	0.450
	*	37	18		26	29		7	48	
Tobacco	No	15	7	0.919	9	13	0.878	2	20	0.697
	Yes	34	15		21	28		6	43	
Alcohol	No	37	29	0.052	20	37	0.024	7	50	0.640
	Yes	12	1		9	4		1	12	

* Tongue/Oral cavity bottom.

**Table 2 ijms-26-06847-t002:** Correlations between TrkA, TrkB and TrkC expression and clinicopathological variables, treating expression as a continuous variable (non-parametric test—Wilcoxon/Mann–Whitney U test).

Feature	TrkA *p*-Value	TrkB *p*-Value	TrkC *p*-Value
Gender	0.177	0.899	0.733
Death	0.299	0.686	0.513
Grade	**0.014**	0.499	0.587
Stage	0.659	0.428	0.601
pT	0.406	0.922	0.625
pN	0.213	0.330	0.382
Localization *	0.784	0.347	0.639

* Tongue/Oral cavity bottom/Other.

**Table 3 ijms-26-06847-t003:** Analyzed clinical and pathological features.

Feature		Number of Cases	%
Gender	Female	31	33.3
	Male	62	66.7
Patient outcome: death	No	69	74.2
	Yes	24	25.8
Grade	1	38	40.9
	2	45	48.4
	3	10	10.7
Stage	I	16	17.2
	II	15	16.1
	III	13	14
	IV	31	33.3
pT	1	16	17.2
	2	24	25.8
	3	14	15
	4	20	21.5
	No data	19	20.4
pN	0	38	40.9
	1	11	11.8
	2	17	18.3
	3	6	6.5
	No data	21	22.6
Surgical treatment	No	21	22.6
	Yes	80	63.4
	No data	13	14
Radiotherapy	No	32	34.4
	Yes	42	45.2
	No data	19	20.4
Chemotherapy	No	65	69.9
	Yes	9	9.7
	No data	19	20.4
Localization of the tumor	Tongue/Oral cavity floor	55	59.1
	Other	38	40.9
Tobacco use	No	22	23.6
	Yes	49	52.7
	No data	22	23.7
Alcohol consumption	No	57	61.3
	Yes	13	14
	No data	23	24.7

## Data Availability

The original contributions presented in this study are included in the article. Further inquiries can be directed to the corresponding author.

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
