# Peer review of "TrkA Expression as a Novel Prognostic Biomarker in Oral Squamous Cell Carcinoma"

_ijms, 2025, doi:10.3390/ijms26146847_

Round 1

Reviewer 1 Report

Comments and Suggestions for Authors

The manuscript presents an investigation into the role of NTRK genes (tropomyosin receptor kinases) and their protein products (Trks) in the pathogenesis and progression of oral cancer. Given the emerging significance of neurotrophic signaling in tumor biology, particularly the involvement of TrkA, TrkB, and TrkC in oncogenic transformation and metastasis, this study addresses an area of increasing interest. The authors aimed to elucidate the expression patterns and potential prognostic value of NTRK genes in oral squamous cell carcinoma (OSCC). They established that TrkA protein correlated with tumor grade and survival (higher TrkA levels- lower grade and better survival). They also found that mRNA expression of NTRK2 and NTRK3 were positively correlated.

Although the study is methodologically limited (qPCR and immunohistochemistry, besides statistical analyses), the topic is relevant, especially in the context of precision oncology, where Trk inhibitors are being evaluated in various solid tumors. Overall, the manuscript has the potential to contribute to the field of oral oncology and molecular pathology.

I have however a number of comments/criticisms:

Title:

Authors should delete the second part of the title: A Comprehensive Analysis of NTRK Gene Family, as the study of NTRK gene family is not  comprehensive (authors did not explore NTRK gene fusions, mutations, etc. and their consequences).

Introduction:

The sentence “DNA methylation changes, histone modifications, non-coding RNA dysregulation, and alterations in the tumor micro-environment and oral microbiome” is unfinished (a verb is missing).

The references 17 and 18 are good but authors should also cite:  DOI: 10.1016/j.ijom.2018.01.020

Authors should explain the relationship between NTRK 1, 2, 3 genes and Trk A,B, C proteins.

Delete the subtitles “Aims of the study” and “Objective” (that are synonymous), and merge the two final paragraphs.

Results

It should be stated at the beginning of this Section how the cut-off values for high and low expression levels for the three proteins (immunostaining) were determined.

Explain the following discrepancy: …when analyzed as a categorical variable (low/high expression), TrkA showed no significant associations with clinicopathological features… In Table 1 I can see association between TrkA and Grade (P=0.024).

Maybe the trend of association between TrkC and pN (P=0.06) should be mentioned.

Correct the Tables’ numbering in the Results and M&M sections, because it is confusing. (probably in an initial version of the Manuscript Table 3 was Table 1?!).

Trk C is missing from Table 2

In Tables 1 and 2, the P values showing significance should be in bold, or with an asterisk.

In the text, authors are referring to Figure 2 for survival analysis, when in fact it is Fig.1. In other words, correct Fig. numbering as well.

I would suggest a graph giving the results of qPCR, i.e. a figure with relative gene expression levels of TRK1, TRK2 and TRK3 in tumor and control tissues.

Materials and methods

Most generally, the methodology is too short. More details are needed.

It is unclear whether RNA was extracted from all the 93 tumor samples?

It is also unclear why palatine tonsils and stomach were used as controls. And what authors meant by “for orientation”?

Why there are no healthy control tissue of the same origin (distant margin for instance). Even a limited number of specimens of healthy oral mucosa should have been useful both in immunohistochemistry and qPCR analyses.

Author Response

We thank You for the work and time committed to improving our manuscript. We did

our best to follow the valuable comments.

  1. Authors should delete the second part of the title: A Comprehensive Analysis of NTRK Gene Family, as the study of NTRK gene family is not  comprehensive (authors did not explore NTRK gene fusions, mutations, etc. and their consequences).

Response to the comment no. 1: The title was changed to: TrkA Expression as a Novel Prognostic Biomarker in Oral Squamous Cell Carcinoma: An Analysis of NTRK Gene Family

  1. The sentence “DNA methylation changes, histone modifications, non-coding RNA dysregulation, and alterations in the tumor micro-environment and oral microbiome” is unfinished (a verb is missing).

Response to the comment no. 2: The sentence was changed to: Key mechanisms include activation of oncogenic pathways (PI3K/AKT/mTOR, MAPK, and Wnt-signaling) [7,15,16], DNA methylation changes, histone modifications, non-coding RNA dysregulation, and alterations in the tumor microenvironment and oral microbiome [16].

  1. The references 17 and 18 are good but authors should also cite:  DOI: 10.1016/j.ijom.2018.01.020

Response to the comment no. 3: The citation was added to the references 17 and 18.

  1. Authors should explain the relationship between NTRK 1, 2, 3 genes and Trk A,B, C proteins.

 Response to the comment no. 4: The information was added in the lines 89-91: ‘NTRK1, NTRK2, and NTRK3 (neurotrophic tropomyosin receptor kinase) refer to a group of genes that encode the TrkA, TrkB, and TrkC proteins, respectively, a family of receptor tyrosine kinases (RTKs).’

  1. Delete the subtitles “Aims of the study” and “Objective” (that are synonymous), and merge the two final paragraphs.

Response to the comment no. 5: The subtitles were changed and the paragraphs were merged.

  1. Results: It should be stated at the beginning of this Section how the cut-off values for high and low expression levels for the three proteins (immunostaining) were determined.

Response to the comment no. 6: The following sentence was added in the section ‘Results’ in the lines 118-120: ‘Assessment of Trk proteins expression was carried applying the H-score approach, which integrates both the intensity of staining (graded on a scale from 0 to 3) and the proportion of positively stained cells (ranging from 0% to 100%).’

  1. Explain the following discrepancy: …when analyzed as a categorical variable (low/high expression), TrkA showed no significant associations with clinicopathological features… In Table 1 I can see association between TrkA and Grade (P=0.024). Maybe the trend of association between TrkC and pN (P=0.06) should be mentioned. Correct the Tables’ numbering in the Results and M&M sections, because it is confusing. (probably in an initial version of the Manuscript Table 3 was Table 1?!). In the text, authors are referring to Figure 2 for survival analysis, when in fact it is Fig.1. In other words, correct Fig. numbering as well.

Response to the comment no. 7: The association between TrkA expression and tumor grade has been included, and the table numbering has been updated accordingly. As noted in lines 123–125: “However, when analyzed as a categorical variable (low/high expression), TrkA showed no significant associations with clinicopathological features except for histological grade (p = 0.024) (Table 1, Table 2).”
Additionally, the observed trend between TrkC expression and pN status has been noted in lines 140–143: ‘Analysis of TrkC expression as a categorical variable showed no significant associations with clinicopathological parameters, although a trend toward an association between TrkC expression and pN status was observed (p = 0.06).’

The numbers of tables and figures have been corrected.

  1. Trk C is missing from Table 2

Response to the comment no. 8: The data has been completed.

  1. In Tables 1 and 2, the P values showing significance should be in bold, or with an asterisk.

Response to the comment no. 9: P values have been changed bold.

  1. I would suggest a graph giving the results of qPCR, i.e. a figure with relative gene expression levels of TRK1, TRK2 and TRK3 in tumor and control tissues.

Response to the comment no. 10: Unfortunately, we did not perform gene expression analysis for NTRK1, NTRK2, and NTRK3 in healthy tissue; therefore, creating such a graph is not possible.

  1. It is unclear whether RNA was extracted from all the 93 tumor samples? Most generally, the methodology is too short. More details are needed.

Response to the comment no. 11: RNA was extracted from the 30 tumor samples, the methodology section has been expanded – lines 293-314: ‘From paraffin blocks of 30 patients with oral cancer, 10 μm x 2 μm sections were cut, which were then subjected to deparaffinization and RNA extraction. Total RNA isola-tion from FFPE sections was performed using a commercial kit—the RNeasy FFPE Mini Kit (QIAGEN), following the manufacturer's instructions.

The concentration of the isolated RNA was determined using fluorometric methods with a Qubit fluorometer (Invitrogen) and the commercial Qubit RNA HS assay kit, in accordance with the manufacturer's protocols.

The mRNA levels of the NTRK1, NTRK2, and NTRK3 genes were determined relative to the transcript level of the reference gene beta-actin (ACTB) using real-time RT-qPCR with TaqMan probes. The RT-qPCR reactions were carried out using the Path-ID™ Multiplex One-Step RT-PCR Kit (Applied Biosystems, Foster City, CA, USA) and Uni-versal ProbeLibrary Probes for Human (Roche) on the LightCycler 480 II system (Roche), with specific primers for both the target and reference transcripts. Each measurement was performed in triplicate, and the final result was reported as the mean of these replicates. The reproducibility of the obtained results was assessed based on the standard deviation and coefficient of variation for each sample. Each RT-PCR reaction included control samples: a negative control (no RNA template) and a positive control, to exclude contamination and confirm the efficiency of the reaction.

The relative expression level of each transcription factor gene was presented as a nor-malized ratio of the expression level of the target transcript to that of the reference gene transcript. Data analysis was performed using the LightCycler 480 II software (Roche Diagnostics International Ltd, Rotkreuz, Switzerland).’

  1. It is also unclear why palatine tonsils and stomach were used as controls. And what authors meant by “for orientation”?

Response to the comment no. 12: When constructing a tissue microarray in a paraffin block, it is standard practice to include tissue cores with a different histological structure than the study material. This allows for the identification of the order of the analyzed cores under the microscope—this is a routine procedure in pathology.

  1. Why there are no healthy control tissue of the same origin (distant margin for instance). Even a limited number of specimens of healthy oral mucosa should have been useful both in immunohistochemistry and qPCR analyses.

Response to the comment no. 13: Unfortunately, the study's ethical approval and informed consent procedures focused on the use of tumor tissue, limiting our access to non-neoplastic tissues for molecular analysis.

We fully agree that including healthy oral mucosa would have enhanced the study by providing a reference for baseline gene and protein expression. We have now added this point to the Limitations section of the manuscript and acknowledge that future studies should aim to incorporate matched normal controls to further validate the prognostic relevance of Trk expression in OSCC.

Reviewer 2 Report

Comments and Suggestions for Authors

General Comments

This article presents a retrospective study on the expression of TrkA, TrkB, and TrkC proteins  in oral squamous cell carcinoma . The authors provide convincing evidence that TrkA is a potential, favorable prognostic biomarker based on immunohistochemical analysis and its correlation with clinicopathological parameters and survival outcomes.

Major Comments

Given the known impact of HPV/EBV on Trk expression in OSCC, their absence in this study is a limitation.Clearly state in the Discussion that HPV/EBV status was not assessed and how this might impact interpretation.

The conclusion that TrkA is a prognostic marker is compelling, but its novelty is somewhat limited in the absence of functional validation. Please consider discussing potential downstream mechanisms.

TrkC expression is high but does not correlate with survival or grade; this requires clearer explanation. Please provide a hypothesis or literature-based explanation for the lack of prognostic value of TrkB/C.

The graph showing IHC staining (Figure 3) could be improved with improved labeling and higher resolution. Please add a scale bar and clarify the staining pattern.

Minor comments

Line 372: “expression of TrkC in OSCC” – incomplete sentence.

Use consistent naming: “TrkA expression” or “NTRK1 protein expression”.

Comments on the Quality of English Language

Repetitive language in the Introduction and Discussion can be streamlined.

Author Response

We express our gratitude for the dedication and effort invested in enhancing our manuscript. We did our best to follow the valuable comments.

  1. Given the known impact of HPV/EBV on Trk expression in OSCC, their absence in this study is a limitation. Clearly state in the Discussion that HPV/EBV status was not assessed and how this might impact interpretation.

Response to the comment no. 1: The information has been added in lines 306-309: ‘An additional limitation regarding the significance of Trk protein expression in our OSCC cohort is the lack of HPV and EBV status assessment in the cancer cells, as these factors can influence tumor biology and potentially affect Trk expression patterns.’

  1. The conclusion that TrkA is a prognostic marker is compelling, but its novelty is somewhat limited in the absence of functional validation. Please consider discussing potential downstream mechanisms.

Response to the comment no. 2: The following text has been added to the discussion in lines 202-213: ‘. TrkA, a high-affinity receptor for nerve growth factor (NGF), is known to activate several key signaling cascades—most notably the PI3K/AKT, RAS/MAPK, and PLCγ pathways—resulting in enhanced cell survival, proliferation, and differentiation [29,30]. These pathways are frequently implicated in head and neck SCCs and could plausibly mediate the observed association between high TrkA expression and im-proved differentiation and survival in our cohort. We now explicitly state that further functional studies, including in vitro and in vivo models, are needed to confirm these mechanistic links and to clarify whether TrkA exerts a causal role in tumor suppres-sion or reflects a more differentiated tumor phenotype [31].

 Further functional studies are needed to explore these regulatory pathways and their implications for tumor biology.’

  1. TrkC expression is high but does not correlate with survival or grade; this requires clearer explanation. Please provide a hypothesis or literature-based explanation for the lack of prognostic value of TrkB/C.

Response to the comment no. 3: The following text has been added to the discussion in lines 174-185: ‘One possible explanation for the limited prognostic value of TrkB and TrkC in our study is their complex and context-dependent biological role. Literature suggests that both TrkB and TrkC can exhibit dual functions depending on the tumor type, stage, or presence of specific ligands. For instance, TrkB has been linked to aggressive behavior and chemoresistance in several cancers due to its role in promoting epithelial-mesenchymal transition (EMT) [24]. As for TrkC, its role is even more ambiguous. While some studies have identified TrkC as a potential tumor suppressor in the absence of its ligand neurotrophin-3 (NT3) [25], other reports show it may support tumor growth under different condition [26]. This dichotomy might explain the lack of asso-ciation with clinical outcomes in our cohort, particularly in a heterogeneous tumor like OSCC, where co-factors such as neurotrophin availability, receptor isoforms, and tu-mor microenvironmental context play a critical role.’

  1. The graph showing IHC staining (Figure 3) could be improved with improved labeling and higher resolution. Please add a scale bar and clarify the staining pattern.

Response to the comment no.4: The resolution of Figure 3 attached to the publication is 72 dpi × 72 dpi (9000 × 3600 pixels). The low quality of the figure included in the manuscript for review was most likely due to the insertion of a preview version. In the final publication, the full-resolution image—identical to the one uploaded during the manuscript submission—will be provided. Scale bars are included in the images, and the figure legend has been updated to describe the staining pattern: lines 287-290: ‘Figure 3. Expression of TrkA in OSCC – membranous and cytoplasmic staining intensity accord-ing to H-score: A. 0, B. 1, C. 2, D. 3,  expression of TrkB in OSCC - membranous and cytoplasmic staining intensity according to H-score: E. 0, F. 1, G. 2, H. 3, expression of TrkC in OSCC - mem-branous and cytoplasmic staining intensity according to H-score: I.  0 , J. 1. (magnification: 10x).’

Round 2

Reviewer 1 Report

Comments and Suggestions for Authors

Although authors have corrected the manuscript to the extent that it was possible (without the additional experiments), I still have one-two minor comments.

As I said in my previous review, the second part of the title should be deleted entirely. Authors deleted only the word "comprehensive"; they did not analyze the NTRK genes actually, but only the gene expression.

In the revised version I did not see the P values in bold.

Also, authors should write uniformly the P values (with 2 or 3 numbers after the period, not as it is now with 1, 2 and 3).

Author Response

We express our gratitude for the dedication and effort invested in enhancing our manuscript. We did our best to follow the valuable comments.

Although authors have corrected the manuscript to the extent that it was possible (without the additional experiments), I still have one-two minor comments.

  1. As I said in my previous review, the second part of the title should be deleted entirely. Authors deleted only the word "comprehensive"; they did not analyze the NTRK genes actually, but only the gene expression.

Response to the comment no.1: The title has been changed to: TrkA Expression as a Novel Prognostic Biomarker in Oral Squamous Cell Carcinoma

  1. In the revised version I did not see the P values in bold. Also, authors should write uniformly the P values (with 2 or 3 numbers after the period, not as it is now with 1, 2 and 3).

Response to the comment no.2: Statistically significant p values were changed bold and p values were rounded to three decimal places throughout.
